

# Study on the relationship between net primary productivity and site quality in Japanese larch plantations in mountainous areas of eastern Liaoning

Wenlong Chang[1], JingHao Li[2], Jinwei Wu[1], Jian Zhang[1], Yang Yu[1], Huiwen Sun[1] and Yibo Wen[1]

[1] Forestry College, Shenyang Agricultural University, Shenyang, China
[2] Center for Biological Disaster Prevention and Control, National Forestry and Grassland Administration, Shenyang, China

Corresponding author
Yibo Wen, wenyibo@syau.edu.cn

## ABSTRACT

Plantation forests enhance carbon storage in terrestrial ecosystems in China. *Larix kaempferi* (Lamb.) Carrière (Lamb.) (*Larix olgensis* Henry) is the main species for afforestation in the eastern Liaoning Province. Therefore, it is important to understand the correlation between the site class and carbon sink potential of *Larix kaempferi* plantations in Liaoning Province for afforestation and carbon sink in this area. The model was fitted using three classical theoretical growth equations: the Richards model, the Korf model, and the Hossfeld model. This study used the forest resource inventory data for management in Liaoning Province in 2011 to build six dynamic height-age models for a *Larix kaempferi* plantation in Dandong City regardless of base-age. The optimal model derived by the generalized algebraic difference approach (GADA) method was compared with the model derived by the algebraic difference approach (ADA) method. The superiority of GADA was demonstrated by comparison. The Levenberg-Marquardt algorithm was used to fit the model. The statistical and biological characteristics were considered synthetically when comparing the models. The best model was screened out by statistical analysis and graphic analysis. The results show that the differential height-age model derived from Richards equation can well explain the growth process of *Larix kaempferi* in Dandong City, Liaoning Province under different conditions. The site index model based on Richards equation and derived by GADA was used to calculate the site class of a *Larix kaempferi* plantation in Dandong City. The net primary productivity (NPP) value from the past ten years was extracted from the MOD17A3HGF data set. Spearman correlation analysis and Kendall correlation analysis were used to show that there is a significant positive correlation between NPP value and site class of *Larix kaempferi* plantation in Dandong City. Among them, the highest growth occurred in 2016; NPP increased by about 3.914 gC/m$^2$/year for every two increases in height-age grade; the lowest increase in NPP was in 2014; NPP increased by about 2.113 gC/m$^2$/year for every two increases in height-age grade; and for every two increases in height-age grade in the recent ten years, the average NPP value increased by about 2.731 gC/m$^2$/year.

## INTRODUCTION

Forests are the largest carbon sequestration system on land with a total biomass of 85%–90%. They have a leading role in regional and global carbon cycles, affecting the cycling of energy, water, and nutrients (*Yao et al., 2021*; *Zhao et al., 2021*; *Morgan et al., 2021*). Further, forests critically influence all carbon stocks, giving them the potential to decelerate or accelerate anthropogenic climate change (*Chen et al., 2023*; *Norbert & Luiza, 2024*). Because of the impacts of climate change and human disturbance, it is important to understand the temporal and spatial changes of forest carbon sources/sinks in order to appropriately plan for regional sustainable development and global carbon emission reduction. Net primary productivity (NPP) refers to the difference between the gross primary productivity (GPP) of an ecosystem and the respiratory energy consumption of vegetation itself. NPP is a key variable to characterize vegetation growth; it is also a direct manifestation of vegetation's own productivity and carbon sequestration capacity (*Wang, 2020*). Site refers to the living space of stand and the natural factors related to the living space. Site quality refers to the productive potential of an established forest or other vegetation types on a certain site (*Yang et al., 2021*). Within the framework of sustainable forest management, measuring site quality and predicting carbon absorption capacity remain a major forestry topic (*Bontemps & Bouriaud, 2014*). The long-term carbon absorption capacity of a forest must be constantly monitored and assessed because of ongoing environmental changes and the introduction of dual-carbon policy (*IPCC et al., 2013*). Forest carbon absorption capacity can be quantitatively evaluated by estimating the NPP value of the ecosystem (*Jeong, You & Hong, 2022*; *Peng et al., 2022*).

As climate change intensifies, the international community is increasingly focusing on the unique carbon sink function of forests. The effective use of carbon absorption capacity has become an important approach for governments to cope with climate change and achieve sustainable economic and social development (*IPCC et al., 2013*; *Guo, 2015*). Research on vegetation NPP started in the 1980s. This field has grown since the implementation of the International Geosphere-Biosphere Program (IGBP) in the 20th century (*Sun et al., 2012*). At present, NPP research methods are mainly divided into two categories: field investigation/observation and model estimation (*Zhou et al., 2022*). Field investigation/observation using the field investigation and eddy covariance technique (EC methods can realize NPP research on an ecosystem scale (*Fang et al., 2007*; *Baldocchi, 2019*). With the increasing importance of carbon sink function and the increasing demand for forest resources, large-scale forest productivity estimation has attracted greater attention. Among them, the model estimation methods suitable for large-scale NPP research can be divided into three categories: the process model (*Wang et al., 2009*; *VEMAP, 1995*), the statistical model, and the light energy utilization model (*Lieth, 1972*; *Lieth, 1975*; *Zhou & Zhang, 1995*; *Prince & Goward, 1995*; *Potter et al., 1993*). The process model has many parameters, which may be difficult to accurately obtain, thus its application is limited. The statistical model is widely used because it is simple, intuitive, easy to implement and requires few parameters. They are often used to describe vegetation on a regional or even global scale (*Zaks et al., 2007*). However, the statistical model considers few climatic factors,
lacks the theoretical basis of physiology and ecology, and lacks universality of statistical laws obtained from different regions and conditions (*Zhou et al., 2022*). At the beginning of the 21st century, with the rapid development of remote sensing technology, large-scale and high-resolution remote sensing data appeared. This provided favorable conditions for quantitatively creating spatial and temporal characteristics of solar energy utilization models. It has become the main method for simulating and estimating large-scale and even global NPP. The Modern Resolution Imaging Spectradiometer (MODIS) is a load device carried on NASA's Earth Observation System satellite. The MODIS sensor is widely used in scientific research and applications because of its high resolution, wide coverage and observation capabilities (*Weng, Fu & Gao, 2014*; *Gao et al., 2015*). Among them, the MOD17A3HGF data set is an NPP product data set obtained from the solar energy utilization model and BIOME-BGC model (biome biogeochemical model). At present, this product has been widely tested and applied in biomass estimation, environmental monitoring, carbon cycle and global change. It reflects the temporal and spatial changes of the NPP value in global ecosystems (*Neumann et al., 2015*; *Kwon & Larsen, 2013*). It has also been used in the NPP study of vegetation in northeast China Therefore, this study uses the MOD17A3HGF data set as the NPP data in the study area (*Mao et al., 2014*; *Running & Zhao, 2021*).

The metabolism of the global ecosystem has been in state of dynamic equilibrium; the carbon cycle had been perpetuated and the carbon storage capacity had remained basically unchanged (*Luo & Xia, 2020*). After entering the industrial society, human activities have accelerated the consumption of fossil resources and increased the emission of greenhouse gases, creating changes in the dynamic equilibrium of the carbon cycle. The global ecosystem has also suffered a series of damages (*Feng et al., 2020*). As an important carbon sink of carbon dioxide ($CO_2$), the terrestrial ecosystem has effectively alleviated global warming (*IPCC et al., 2013*). Forests are a significant ecosystem that play an important role as a carbon sink (*Wang, Shi & Hu, 2022*). According to the data of the Ninth National Forest Inventory (*China National Forestry and Grassland Science Data Center , 2018*), China's forest area, planted area, and timber volume rank the first in the world. To some extent, secondary forests can buffer climate change (*Frelich et al., 2020*). At present, China's forests are young growths and half-mature forests. Many studies show that young growths and half-mature forests have higher carbon sink potential (*Piao et al., 2022*). The site index calculated by the mathematical model based on historical growth data and observation data represents the predicted growth potential of specific tree species or stand planted in a site and can provide information about suitable tree species in a specific site and predict their growth rate and potential (*Socha et al., 2020*; *Rahimzadeh-Bajgiran et al., 2020*). The site index (SI) is the commonly used value to measure site productivity, that is, the height of dominant trees at index age. The statistical model describing the relationship between dominant tree height and index age is called the SI model. An SI model can transform the dominant tree height of real stand into the tree height at exponential age. The ADA is one of the main methods used to construct the SI model (*Bailey & Clutter, 1974*), however, this method can only construct a series of unicursal curves with multiple horizontal asymptotic extremes or a series of polymorphic curves with only one horizontal

asymptotic extreme (*Cao & Sun, 2017*). To solve this problem, Cieszewski and others put forward generalized algebraic difference approach (GADA) (*Cieszewski & Bailey, 2000*). Since GADA can construct polymorphic SI curve with an asymptomatic variable horizontal extreme, it has received widespread attention and has become the main method to construct the SI model in recent years (*Anta et al., 2011*; *Castillo-Lopez et al., 2018*; *Adan et al., 2022*). Some scholars have developed new mixed models based on the ADA method, which have lower data collection and calculation costs (*Protazio et al., 2022*). In addition, *López-Álvarez, Franco-Vázquez & Marey-Perez (2023)* expanded the application scope of the GADA method. They use the GADA model to predict the annual cumulative resin output of *Pinus pinaster* Ait. However, the modeling data used by GADA to build the SI model mainly include the stem analysis data of dominant trees and the inventory data of fixed sample plots (*Souza et al., 2022*; *Mfilho et al., 2023*). In practical application, it is difficult to obtain the stem analysis data, which requires destructive sampling and has a high technical threshold. As the survey data of forest resources planning and design, forest resource inventory data for management is more macroscopic and more suitable to be combined with remote sensing data in a larger scale. In this study, the SI model is used to establish the model and using the data obtained from the average tree height. Therefore, the model is the average tree height-age model, and the average tree height has a strong correlation with land productivity (*Guo et al., 2023*). The height-age model was used to approximately represent the site quality. According to the average height-age model, the site quality grade was divided and resulted in the average height-age quality (H-AQ).

*Larix kaempferi* (Lamb.) Carrière (Lamb.) (*Larix olgensis* Henry) is the main species for afforestation in eastern Liaoning. Therefore, understanding the correlation between the site class and carbon sink potential of *Larix kaempferi* plantations in the Liaoning Province is important for afforestation purposes and to predict carbon sink in this area.

## MATERIALS & METHODS

### Study area

Dandong City was used as the study area (Fig. 1). Dandong City is located in the center of northeast Asia, with a geographical position of 123 23′–125 42′ east longitude and 39 44′–41 09′ north latitude. It is adjacent to the confluence of Yalu River and Yellow Sea and is an important area in the southeast of Liaodong Peninsula. It is the warmest and wettest place in Northeast China with a temperate sub-humid monsoon climate. The annual average rainfall is between 800–1,200 mm. Affected by monsoon, seasonal changes are obvious with four distinct seasons.

### Sources of data

#### Sample plot

The data of this study comes from the forest resource inventory data for management in Liaoning Province in 2011. This study is a continuation and strengthening of the fixed-point survey and monitoring of forest area, forest species, and forest age in the Second National Land Survey. The contents of the survey include forest area, storage capacity, timber volume, stand structure, and ecological function. In the forest resource inventory data for

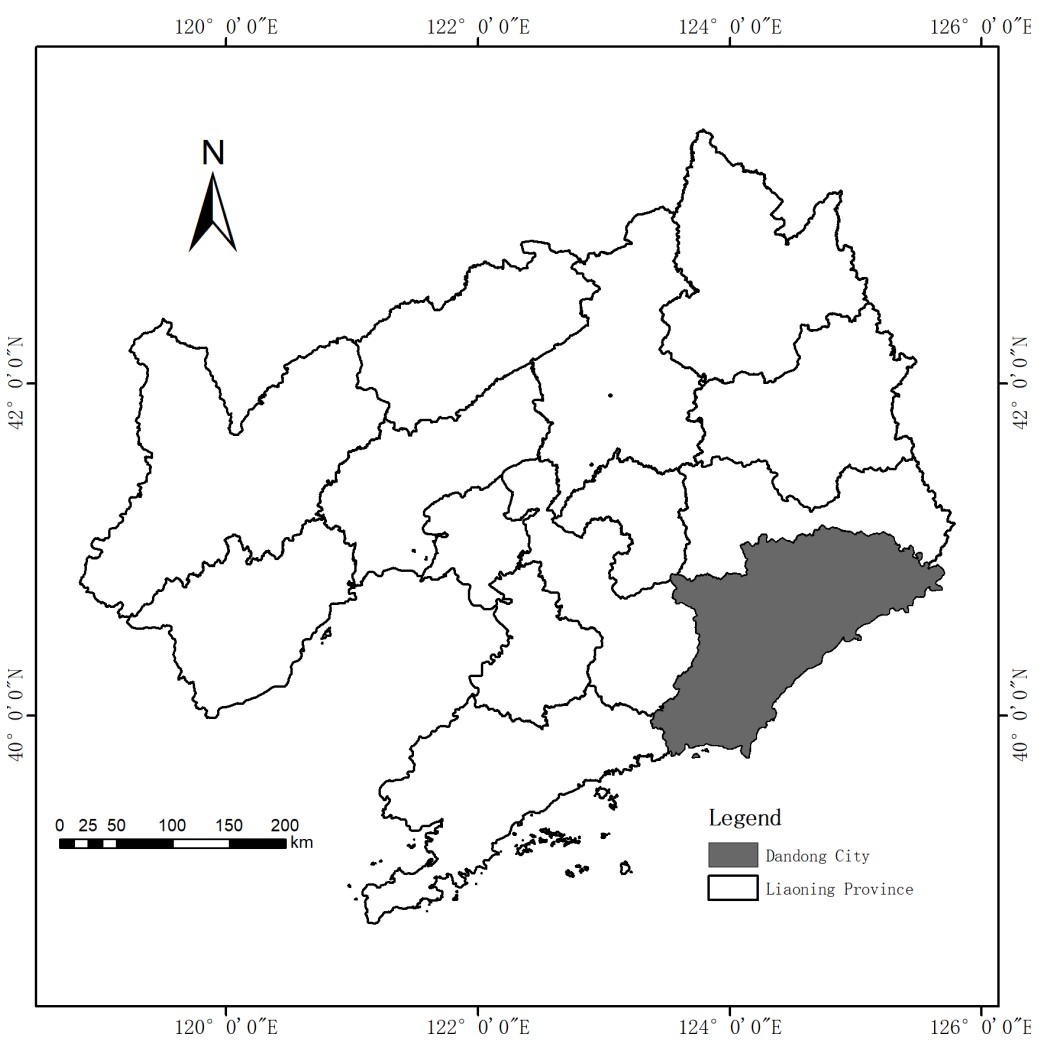

**Figure 1** Location map of the study area: Dandong City (grey).

management, 7,077 sample plots with an area greater than 2 hm² and a canopy density greater than 0.2 were selected for research. Figure 2 shows the shape of the woodland. Most of the growth in the 7,077 sample plots are young growths and half-mature forests. Among them, 6,811 sample plots have an average age of less than 40 years, accounting for 96.2% of the total. The average height of each age class ranges from 11.5 m to 24.7 m (see Table 1).

To generate the family of H-AQ curves, pairs of observations of tree height and age of all plots with at least two measurements were fitted to four dynamic equations expressed under the GADA technique. We used the 2005 Liaoning Provincial Forest Resources Survey Data to match the 2011 Liaoning Provincial Forest Resources Survey Data to obtain 5,601 sets of data for modeling. Tree ages spanned from 0~70 years and were divided into seven age intervals according to an age interval of 10 years. See Table 2 for stand characteristics of sample sites.

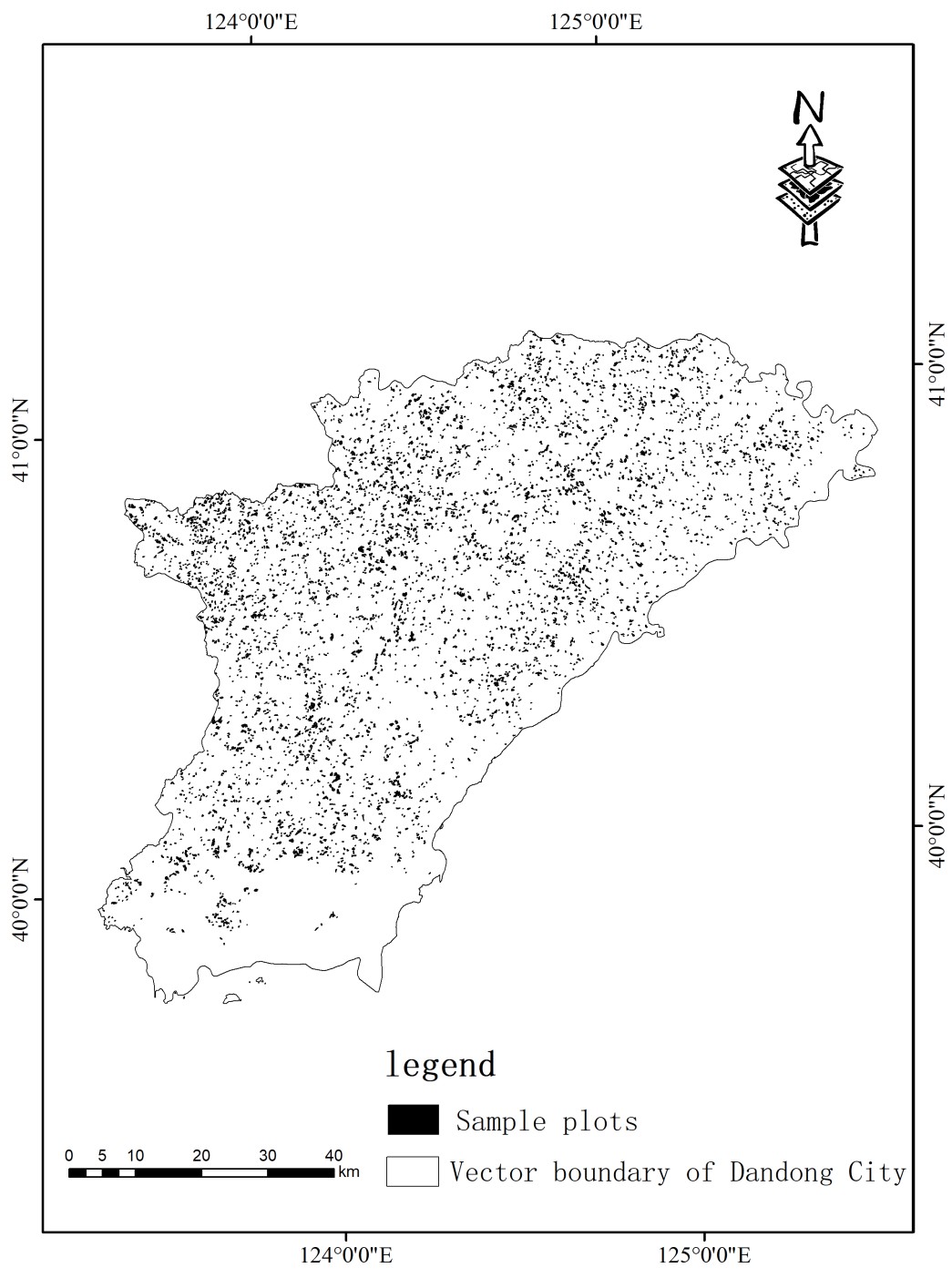

**Figure 2** **Distribution map of the Larix kaempferi plantation in Dandong City.** The black area is the sample plots.

**Table 1  Statistics of age distribution of the sample plots.** Description of forest class characteristics based on forest survey data in 2011.

| Age class/a | Sample plot number/N | Age/a | Average height/m |
|---|---|---|---|
| Young growth | 3,590 | 0~20 | 11.5 |
| Half-mature forest | 2,460 | 21~30 | 14.8 |
| Nearly mature forest | 760 | 31~40 | 17.2 |
| Mature forest | 215 | 41~60 | 19.2 |
| Overmature forest | 52 | >60 | 24.7 |

**Table 2  Statistics of age distribution of the sample plots.** The table summarizes the characteristics of the data used for modeling, including the average tree height, maximum tree height, minimum tree height, and the number of stands in each age group for the years 2005 and 2011.

| Year | Age interval/a | Sample plot number/N | Standard deviation/m | Minimum value/m | Maximum value/m | Average tree height/m |
|---|---|---|---|---|---|---|
| 2005 | 0~10 | 1,199 | 2.2 | 0.9 | 13.0 | 5.5 |
| 2005 | 11~20 | 2,754 | 2.4 | 3.0 | 19.5 | 10.1 |
| 2005 | 21~30 | 1,285 | 2.7 | 4.5 | 24.0 | 14.0 |
| 2005 | 31~40 | 246 | 2.8 | 7.4 | 25.3 | 17.1 |
| 2005 | 41~50 | 64 | 3.3 | 11.5 | 25.0 | 18.2 |
| 2005 | 51~60 | 23 | 4.6 | 10.0 | 29.5 | 21.5 |
| 2005 | >60 | 35 | 3.0 | 18.3 | 29.9 | 24.9 |
| 2011 | 0~10 | 66 | 1.2 | 2.4 | 7.3 | 3.7 |
| 2011 | 11~20 | 2,518 | 3.0 | 3.1 | 22.7 | 12.2 |
| 2011 | 21~30 | 2,102 | 3.0 | 6.8 | 24.3 | 14.9 |
| 2011 | 31~40 | 669 | 3.3 | 5.1 | 27.2 | 17.0 |
| 2011 | 41~50 | 164 | 3.1 | 8.4 | 27.5 | 19.26 |
| 2011 | 51~60 | 30 | 3.0 | 12.9 | 26.4 | 18.85 |
| 2011 | >60 | 52 | 4.1 | 10.2 | 31.1 | 24.9 |

## MODIS NPP

This study uses NPP remote sensing image provided by the NASA website (https://urs.earthdata.nasa.gov/). It is the product of MOD17A3HGF which is the MODIS image with spatial resolution of 500 m. NPP was calculated by determining the net photosynthesis (PSN), *i.e.,* the difference between gross primary productivity and maintenance respiration costs (MR) every 8 days. The original format of the data is HDF, and the temporal resolution is annual. The data uses MODIS/TERRA satellite remote sensing parameters and simulates the NPP data set through the BIOME-BGC ecosystem model.

The downloaded data in HDF format were spliced, resampled, and reprojected in batches using the MRT tool. These data were multiplied by the scale factor of 0.1, and the attribute value representing non-vegetation pixels was assigned as a null value. Finally, based on the vector data of the study area, the remote sensing data of NPP time series of vegetation in Dandong City from 2013 to 2022 were obtained in batches.

## Research methods
### Selection of baseline model

Three methods are typically used to construct the SI curve: the guide curve method, the parameter estimation algorithm, and ADA. Among them (*Barrio, 2005*), ADA has gradually become the preferred method for site quality curve fitting (*Duan & Zhang, 2004*). At present, there are three derivation methods of differential SI models (*Zhao, Ni & Gordon, 2012*): (1) the derivative integration method proposed by *Clutter (1963)*; (2) the algebraic difference approach (ADA) proposed by *Bailey & Clutter (1974)*; (3) GADA proposed by *Cieszewski & Bailey (2000)*. When deducing with these three methods, "site-dependent parameter (SDP)" and "site-independent parameter (SIP)" must be specified in advance in the basic model (*Barrio, 2005*). Compared with the derivative integration method, ADA and GADA are relatively simple and are widely used. For the SI model, the following most important and ideal properties are needed: pleomorphism; invariance of reference age; S-shaped growth curve with an inflection point passing through the origin of coordinates (when the age is 0, the tree height should be 0); and multiple horizontal asymptotes (equation curves do not decrease with age). The SI model derived from the basic equation and constructed by GADA has excellent properties and can meet these conditions. GADA specifies several model parameters as site-related parameters, *i.e.,* parameter related to site, and derives dynamic GADA formulation with variable horizontal asymptotes, so that site quality can be accurately and effectively predicted. Three growth equations, namely the Richards, Korf, and Hossfeld equations, are widely used in the research of forest and stand growth models. Here, these three equations are used as the basic equations for deriving the differential H-AQ model by GADA.

Richards equation:

$$h = a\left(1 - e^{-bt}\right)^c. \tag{1}$$

Lundqvist-Kolf equation:

$$h = ae^{-bt^{-c}}. \tag{2}$$

Hossfeld Equation:

$$h = \frac{a}{1 + bt^{-c}}. \tag{3}$$

In the equations: a is the limit value parameter of tree height (potential maximum tree height) representing the maximum value of tree height under certain site conditions; b is the growth rate parameter (or scale parameter) and affects the inflection point of the curve; c is the shape parameter and affects the shape and inflection point of the curve (*Niu, Dong & Li, 2020*).

### Derivation of the difference model

In 1974, Bailey and others put forward ADA. Based on this Cieszewski suggested GADA (*Cieszewski & Bailey, 2000*). GADA can be applied to several site-related parameters to derive polymorphic dynamic GADA formulation with variable horizontal asymptotic lines, so as to quickly and accurately predict site quality and build SI models (*Cieszewski, 2001*; *Cieszewski, 2002*).

Derivation of the differential SI model by GADA usually includes the following steps:

(1) Select a growth equation as the basic equation.

(2) Specify two or more parameters in the equation as parameters related to site.

(3) Propose a variable $X_0$ related to site quality. Assume that parameters related to site have various quantitative relationships with the variable (such as linear, inverse function, quadratic, exponential, *etc.*).

(4) Substitute the above functional into the basic equation and solve $X_0$, here defined as $(t_0, h_0)$. Among them, $t_0$ is the specified forest age, $h_0$ is the tree height under the specified forest age. When $t_0$ is the benchmark forest age, $h_0$ is the SI.

(5) Substitute the solved $X_0$ into the basic equation, and the differential SI derived by GADA can be obtained.

The SI model constructed by this method can satisfy two attributes: multiple horizontal asymptotes and polymorphism. Taking Eq. (1) as an example, the parameters a and b in Eq. (1) are set as parameters related to site. Then set a linear relationship between parameter a and $X_0$, and an inverse proportional function relationship between parameter b and $X_0$. That means $a = e^{X_0}$, $c = c_1 + c_2 X_0$. Equation (1) can be transformed into:

$$h = e^{X_0} \left(1 - e^{-bt}\right)^{(c_1 + c_2 X_0)}. \tag{4}$$

Then the following can be derived:

$$X_0 = \frac{\ln h - c_1 \ln\left(1 - e^{-bt}\right)}{1 + c_2 \ln\left(1 - e^{-bt}\right)}. \tag{5}$$

In Eq. (5), $t = t_0$, $h = h_0$, of which $t_0$ is the reference age and $h_0$ the tree height at the specified age. When $t_0$ is the reference age, $h_0$ will be the SI. Make Eq. (4) be $t = t_1$, $h = h_1$, of which $t_1$ is the predict age and $h_1$ the tree height under that age. Substitute Eq. (5) into Eq. (4), and the differential SI model based on Eq. (3) can be obtained by GADA. The representation is:

$$h_1 = e^{X_0} \left(1 - e^{-bt_1}\right)^{(c_1 + c_2 X_0)}. \tag{6}$$

Among them, $X_0$:

$$X_0 = \frac{\ln h_0 - c_1 \ln\left(1 - e^{-bt_0}\right)}{1 + c_2 \ln\left(1 - e^{-bt_0}\right)}. \tag{7}$$

In order to compare the difference between GADA and ADA, based on Richards equation, difference model E0 can be derived by ADA (Table 3).

The methods SI model was used, and the average tree height was used as the data. The final model was the mean height -age model.

## Model fitting and validation

The model parameter estimation uses the nlsLM function in the minpack.lm package of R language. This function uses the Levenberg–Marquardt algorithm to solve the nonlinear least squares problem (*Arias-Rodil et al., 2014*). There are two main aspects for testing the goodness of the model: One is the biological meaning of the model and the parameters,

**Table 3 Difference site index equations and base equations.** E0 is ADA model as control, and E1, E2, E3 and E4 are GADA models.

| Base equation | Parameter related to site | Solution for $X$ | Dynamic GADA/ADA formulation | No. |
|---|---|---|---|---|
| | $a = X_0$ | $X_0 = h_0 (1 - e^{-bt_0})^{-c}$ | $h_1 = h_0 \frac{(1-e^{-bt_1})^c}{(1-e^{-bt_0})^c}$ | E0 |
| Richards: $h = a(1 - e^{-bt})^c$ | $a = e^{X_0}$ $c = b_1 + 1/X_0$ | $X_0 = 1/2(\ln h_0 - b_1 F + \sqrt{(b_1 F - \ln h_0)^2 - 4F})$ $F = \ln(1 - e^{-bt_0})$ | $h_1 = e^{X_0}(1 - e^{-bt_1})^{(b_1 + 1/X_0)}$ | E1 |
| | $a = e^{X_0}$ $c = c_1 + c_2 X_0$ | $X_0 = \frac{\ln h_0 - c_1 \ln(1 - e^{-bt_0})}{1 + c_2 \ln(1 - e^{-bt_0})}$ | $h_1 = e^{X_0}(1 - e^{-bt_1})^{(c_1 + c_2 X_0)}$ | E2 |
| Lundqvist-Kolf: $h = a e^{-bt^{-c}}$ | $a = e^{X_0}$ $b = b_1 + b_2/X_0$ | $X_0 = \frac{1}{2}(b_1 t_0^{-c} + \ln h_0 + F)$ $F = \sqrt{(b_1 t_0^{-c} + \ln h_0)^2 + 4b_2 t_0^{-c}}$ | $h_1 = e^{X_0} e^{(-(b_1 + b_2/X_0 t_1^{-c}))}$ | E3 |
| Hossfeld: $h = a/(1 + bt^{-c})$ | $a = b_1 + X_0$ $b = b_2 X_0$ | $X_0 = \frac{h_0 - b_1}{1 + b_2 h_0 t_0^{-c}}$ | $h_1 = \frac{b_1 + X_0}{1 + b_2 X_0 t_1^{-c}}$ | E4 |

and the other is the actual fitting effect of the model characterized by statistical indicators. This study uses three statistical indicators commonly used in regression analysis: the determination coefficient ($R^2$), bias (BIAS), root mean square error (RMSE). Fitting samples are used to calculate the above three indicators; the closer the determination coefficient is to 1, the smaller the bias, the smaller the root mean square error and the smaller the mean absolute error, the better the prediction effect of the model. The formulas of indicators are as follows:

$$R^2 = 1 - \frac{\sum_{i=1}^{n}(h_i - \hat{h}_i)^2}{\sum_{i=1}^{n}(h_i - \overline{h})^2} \tag{8}$$

$$BIAS = \frac{\sum_{i=1}^{n}(h_i - \hat{h}_i)}{n} \tag{9}$$

$$RMSE = \sqrt{\frac{\sum_{i=1}^{n}(h_i - \hat{h}_i)^2}{n}} \tag{10}$$

in which: $h_i$ is the measured value of the tree height; $\hat{h}_i$ is the estimated value the tree height; $\overline{h}$ is the average of the measured values of tree height; n is the number of samples.

According to the above indicators, several models were evaluated and the best fitting model of the height-age curve cluster was chosen to verify whether the model met the properties of an ideal height-age model. All statistical analyses above were performed in R4.2.3 using R Studio (*R Core Team, 2022*; *RStudio Team, 2002*).

## Selection of base age and H-AQ class interval

The tree heights of stands should be stable at the reference age and there should be obvious differences in the tree heights in different site conditions. There is no obvious influence of tree age on the site quality evaluation results of most tree species (*Clutter, Fortson &*

*Pienaar, 1983*). Generally, the following aspects are considered: (1) the age class after the growth tends to be stable; (2) cutting age; (3) half of the natural maturity age; (4) the age when the average volume or tree height is maximum, *etc.* (*Guo, Zhang & Zhang, 2007*). However, China typically uses 1 m or 2 m as the SI interval and the number of index levels rarely exceeds 10 (*Meng & Chen, 2001*). Height-age grades also adopt this method. Therefore, combined with sample data, we set the reference age to 30 years ($t_b$), the H-AQ class interval 2 m, and the range of H-AQ from 10 m to 26 m.

## NPP resolution of forest land under different site quality

Resolution (R) is an indicator to measure the degree of data separation. In this study, we use R to calculate whether the NPP value shows a separation trend under different site qualities and evaluate the influence of different site qualities on NPP value. If the result of R is positive, it indicates that the absolute data increases, and vice versa. The greater the absolute value, the higher the degree of data separation. Here, the influence of site quality on NPP of forest land was analyzed by calculating the NPP resolution of forest land with different site qualities. Its calculation formula is as follows:

$$R = 2 * \frac{T_2 - T_1}{W_{B1} + W_{B2}} \tag{11}$$

in which: T is the average NPP, and $W_B$ is 2.354 times of NPP standard deviation. The numbers 1 and 2 represent different site qualities.

## Correlation analysis between H-AQ and NPP

To explore the changes of NPP under different site categories, we drew a box diagram of the average NPP in the past ten years and made a linear regression analysis between the average NPP in the past ten years and H-AQ. Spearman correlation analysis and Kendall correlation analysis were used to deeply study the correlation between H-AQ and NPP. All the above tests and analyses were carried out in *OriginLab, (2021)*.

The exponential order derived by GADA was divided into three levels to classify site quality. Specifically, lands with an exponential order of 10∼14 are classified as inferior sites, lands with 16∼20 as average-quality sites, and lands with 22∼26 as high-quality sites.

To explore the relationship between age class and NPP, we selected the data of 2013 for analysis, which was divided into an age class every ten years with 0–10 years as Grade I, 10–20 years as Grade II, and so on. Because there are few trees over 50 years old, trees over 50 years old were classified as Grade VI.

# RESULTS

## Model fitting results
### Model selection and evaluation
Modeling samples from a *Larix kaempferi* (Lamb.) Carrière (Lamb.) plantation (Table 2) were used to fit each differential H-AQ equation (Table 3). The results of parameter estimation, *P*-value and determination coefficient ($R^2$), bias (BIAS), root mean square error (RMSE) are shown in Table 4.

**Table 4 Parameter estimates and goodness of fit of the difference height-age equations for *Larix kaempferi* (Lamb.) Carrière (Lamb.) Carr.**

| Model | Parameter | Estimation | $p$-value | $R^2$ | BIAS | RMSE |
|---|---|---|---|---|---|---|
| E0 | $b$ | 0.0495215 | $<0.0001$*** | 0.9702468 | 0.02548458 | 0.6806003 |
|  | $c$ | 1.4466216 | $<0.0001$*** |  |  |  |
| E1 | $b$ | 0.0507173 | $<0.0001$*** | 0.9707875 | 0.01423038 | 0.6743881 |
|  | $b_1$ | 1.1624718 | $<0.0001$*** |  |  |  |
| E2 | $b$ | 0.0518524 | $<0.0001$*** | 0.9709394 | 0.005271001 | 0.6726317 |
|  | $c_1$ | 2.0926676 | $<0.0001$*** |  |  |  |
|  | $c_2$ | $-0.1844717$ | $<0.0001$*** |  |  |  |
| E3 | $b_1$ | 2.326745 | 0.0003 | 0.9635747 | $-0.03073686$ | 0.7530557 |
|  | $b_2$ | 22.538452 | $<0.0001$*** |  |  |  |
|  | $c$ | 0.373826 | $<0.0001$*** |  |  |  |
| E4 | $b_1$ | 66.842234 | $<0.0001$*** | 0.9671859 | $-0.03708711$ | 0.7147521 |
|  | $b_2$ | 0.104840 | $<0.0001$*** |  |  |  |
|  | $c$ | 1.263434 | $<0.0001$*** |  |  |  |

**Notes.**
***Indicates that the significance level is extremely significant.
$R^2$ indicates the coefficient of determination. BIAS indicates the bias. RMSE indicates the root mean square error.

In the standard of model fitting, the closer the R2 value is to 1, the stronger the explanatory power of the model to the actual data, and the smaller the BIAS and RMSE, the smaller the difference between the predicted results of the model and the real observed values, and the higher the prediction accuracy. Graphically, it is better for the model's growth curve to be more in line with the actual growth of trees. The above two conditions were considered when selecting the model.

Finally, several models were evaluated according to all the above indicators and the best fitting model of the height-age curve cluster was used to verify whether the model met the properties of an ideal height-age model. All statistical analyses above were performed in R4.2.3 using R Studio.

By comparison, it can be seen from Table 4 that the parameter estimation of the four models was highly significant ($<0.01$). All of the $R^2$ are above 0.96, which shows that the four models have good fitting effect for modeling data. Among them, the BIAS value of E2 model is closer to 0, which means that the prediction BIAS of E2 model is smaller, the dispersion is lower, and the root mean square error (RMSE) is below 0.76. The RMSE of E2 is the smallest and therefore the fitting effect is the best, followed by the E2 model derived by GADA. These results show that all the above models have good forecasting ability. Considering the various indicators, the E2 model was selected for further testing.

### Comparison of models derived from GADA and ADA

To reveal the advantages of GADA, we selected the E0 model based on the same Richards equation using ADA for subsequent comparative analysis. The H-AQ curve clusters of E0 and E2 were drawn (Fig. 3) and a comparative graph was created for the three site index curve clusters (10 m, 18 m, 26 m) of the two models with good or bad performance (Fig. 4).

The growth processes of the trees in the two models are basically the same under average site conditions (18 m; Fig. 4). In trees over 35 years, the curve at the minimum H-AQ of
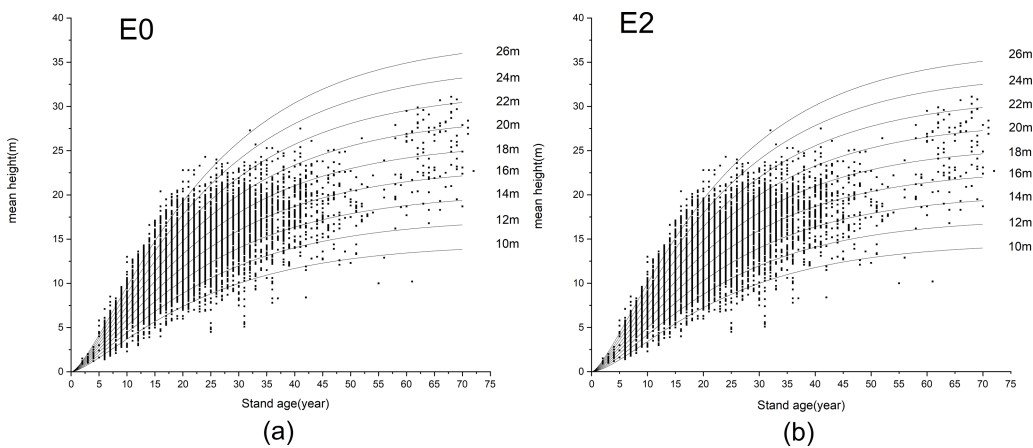

**Figure 3  Site index curves generated with the difference mean height-age models E0 and E2, (A) model E0; (B) model E2.** Scattered points in the figure show the relationship between age and tree height of stands in 2011. 10 m–26 m is H-AQ grade, H-AQ stands for the average height-age quality.

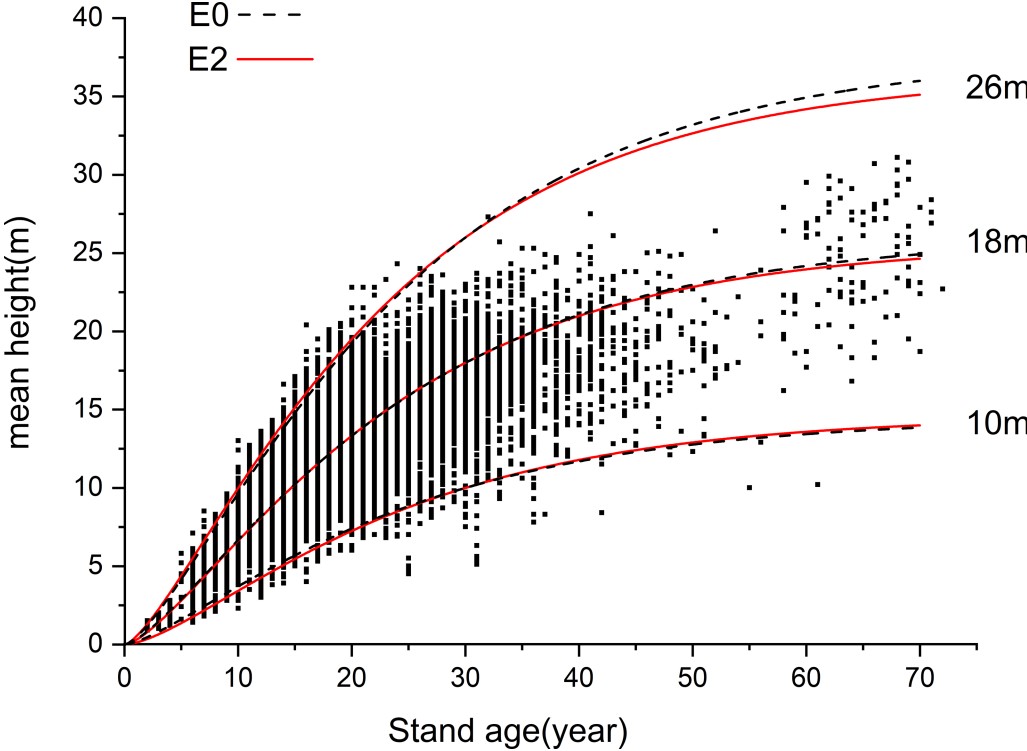

**Figure 4  Comparison of difference mean height-age models E0 and E2.** The black line segment is the growth curve of model E0 when the H-AQ is 10, 18 and 26, and the red curve is the growth curve of model E2 when the H-AQ is 10, 18 and 26. Scattered points in the figure show the relationship between age and tree height of stands in 2011.

**Table 5  Comparison between model E0 and model E2 in terms of inflection, annual maximum height growth rate and dominant tree height at influence.** The inflection point of the curve is at the maximum slope.

| H-AQ/m | E0 | | | E2 | | |
|---|---|---|---|---|---|---|
| | Inflection/ year | Annual maximum height growth rate/m | Dominant tree height at inflection/m | Inflection/ year | Annual maximum height growth rate/m | Dominant tree height at inflection/m |
| 10 | 7.4 | 0.4246 | 2.6238 | 9 | 0.4207 | 3.0196 |
| 12 | 7.4 | 0.5096 | 3.1486 | 8.6 | 0.5070 | 3.5093 |
| 14 | 7.4 | 0.5945 | 3.6734 | 8.3 | 0.5940 | 4.0045 |
| 16 | 7.4 | 0.6794 | 4.1982 | 8.0 | 0.6816 | 4.4609 |
| 18 | 7.4 | 0.7644 | 4.7229 | 7.7 | 0.7698 | 4.8764 |
| 20 | 7.4 | 0.8493 | 5.2477 | 7.5 | 0.8587 | 5.3354 |
| 22 | 7.4 | 0.9343 | 5.7725 | 7.2 | 0.9482 | 5.6737 |
| 24 | 7.4 | 1.0192 | 6.2973 | 7.0 | 1.0382 | 6.0709 |
| 26 | 7.4 | 1.1041 | 6.8220 | 6.8 | 1.1289 | 6.4402 |

model E2 was above model E0, while the curve at the maximum H-AQ was below model E0. From the scattergram, it can be observed that the height growth of *Larix kaempferi* plantations is slow when the age is less than 5. However, after the age of 5, the height growth showed an obvious upward trend. At this point, model E2 shows a slow upward trend first and then a rapid upward trend. In trees over 35 years, the curve at the minimum H-AQ of model E2 was below model E0, while the curve at the maximum H-AQ was above model E0; model E2 contains more data points. This indicates that in younger trees, model E2 better fits the growth process of *Larix kaempferi* in Dandong under different site conditions. Compared with ADA, the equations derived by GADA have two properties: multiple horizontal asymptotes and pleomorphism. To further verify this, we compared the inflection point position, the slope at the inflection point (the maximum annual growth) and the function value at the inflection point (the tree height at the inflection point) of the two curves to further illustrate the differences between the two methods after drawing the H-AQ curve clusters of E0 and E2. If a group of curves satisfies multiple horizontal asymptotes, the curves under different H-AQ will have different maxima, that is, different asymptotes; if a group of curves satisfies pleomorphism, the curves under different H-AQ will have different inflection points, and there should be no simple proportional relationship between curves. It can be seen in Table 5 that the differential H-AQ model E0 obtained by ADA only has a simple proportional relationship between the extreme values of curves under different H-AQ; their inflection points are exactly the same, while the slope at the inflection points is also proportional to the predicted values. This indicates that it satisfies the properties of multiple horizontal asymptotes but does not satisfy pleomorphism. In the differential H-AQ model E2 established by GADA, with the increase of H-AQ, the difference of extreme value between adjacent curves decreases gradually, the position of inflection points shifts obviously to the left, and the difference of slope at inflection point of adjacent curves increases gradually; at the same time, it satisfies the properties of multiple horizontal asymptotes and pleomorphism. Therefore, it can better predict the growth process of *Larix kaempferi* (Lamb.) plantation under different site conditions.

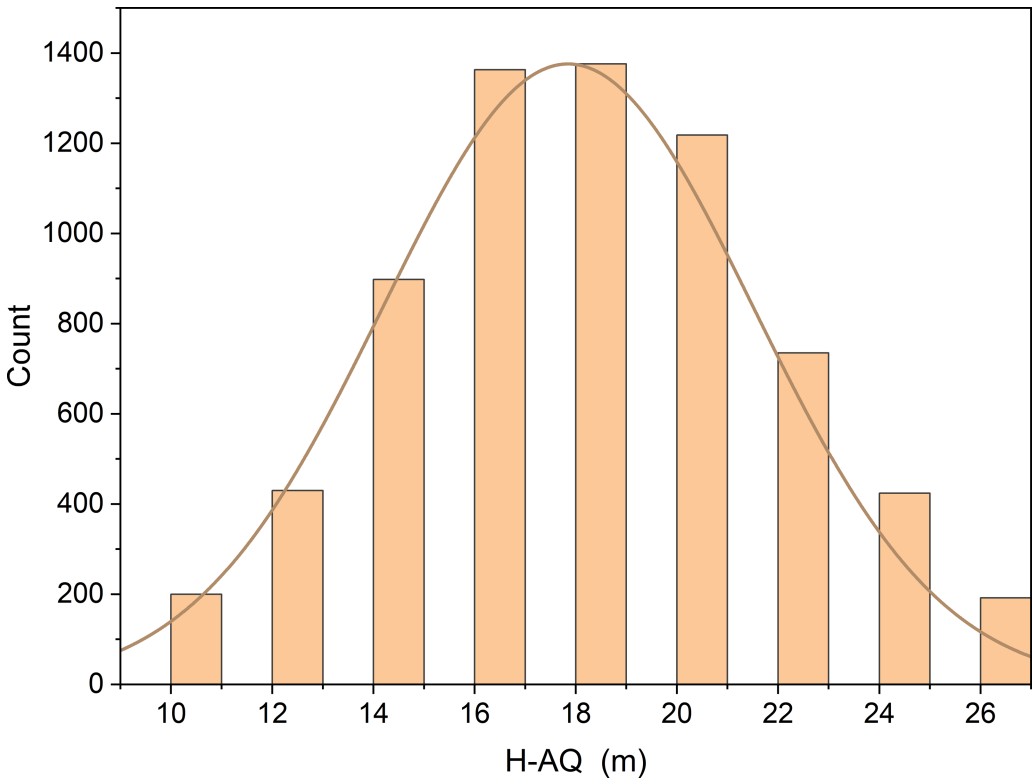

**Figure 5** **Number distribution of each site class.** Grading site quality: height-age quality (H-AQ), and the unit of H-AQ is the meter.

### Site quality of a *Larix kaempferi* (Lamb.) Carrière (Lamb.) plantation in Dandong City

The area of forest land in each site was calculated and the results are shown in Fig. 5. The NPP values of sample plots in recent ten years was extracted by ArcMap 10.8 (the data comes from MOD17A3HGF-NPP data set), and its spatial distribution pattern is shown in Fig. 6. Forest land was divided into three site types according to status grade. This classification can evaluate ecological environment, resource utilization and productivity level according to site quality levels and provide reference for rational management and utilization of forest land resources. At the same time, this classification also meets the actual needs; it is simple, feasible, and has wide applicability. Among them, inferior sites account for 22%, average-quality site 58%, and high-quality sites 20%.

Figure 6 shows the spatial distribution of *Larix kaempferi* 's site quality in Dandong City. It can be seen from the map that the compartments with better site qualities are mainly located in the west and north, showing a spatial distribution of regular dispersion and partial aggregation.

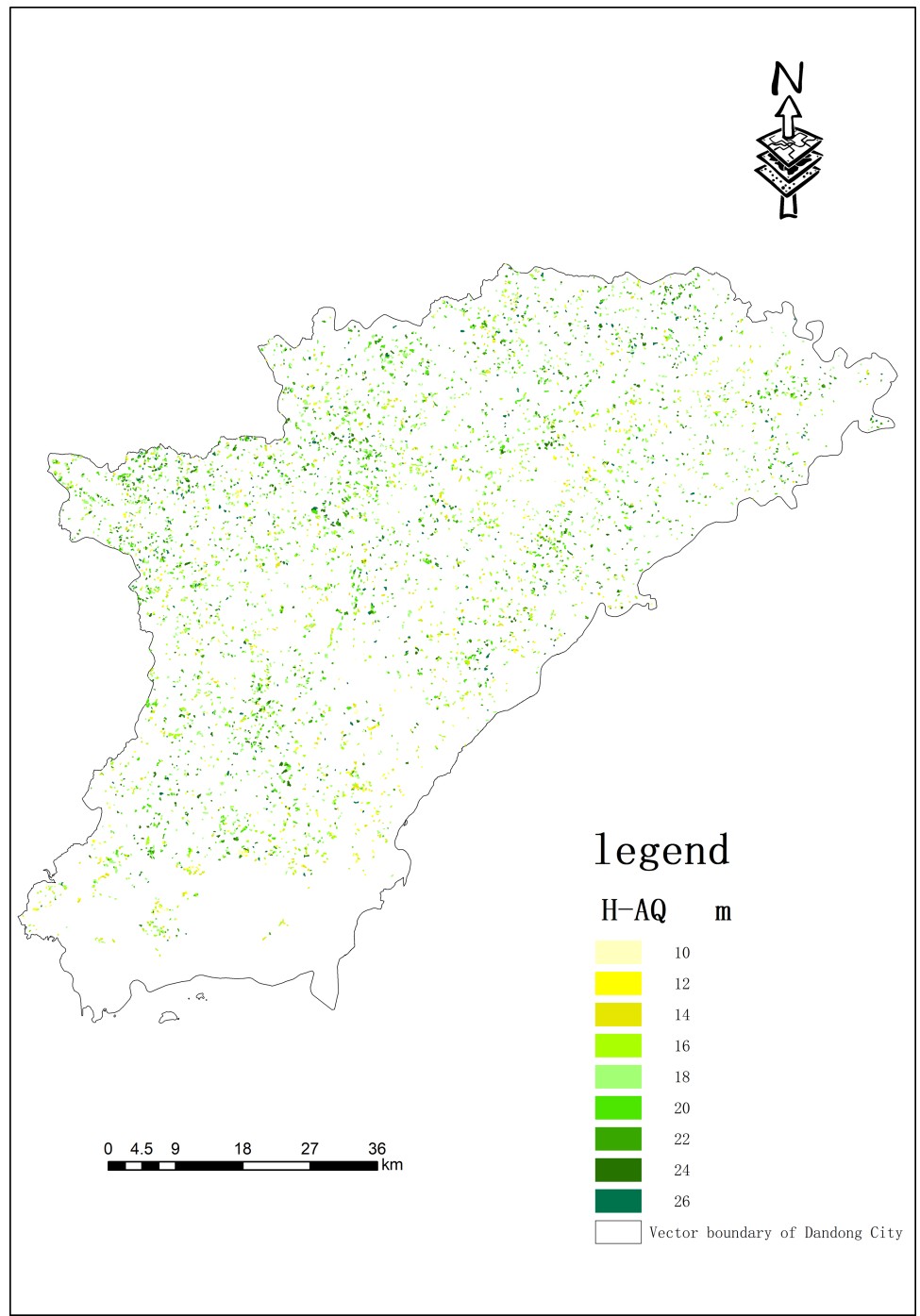

**Figure 6   Spatial distribution of site qualities of *Larix kaempferi* (Lamb.) Carrire (Lamb.) com part-ments in Dandong City.** H-AQ stands for the average height-age quality.

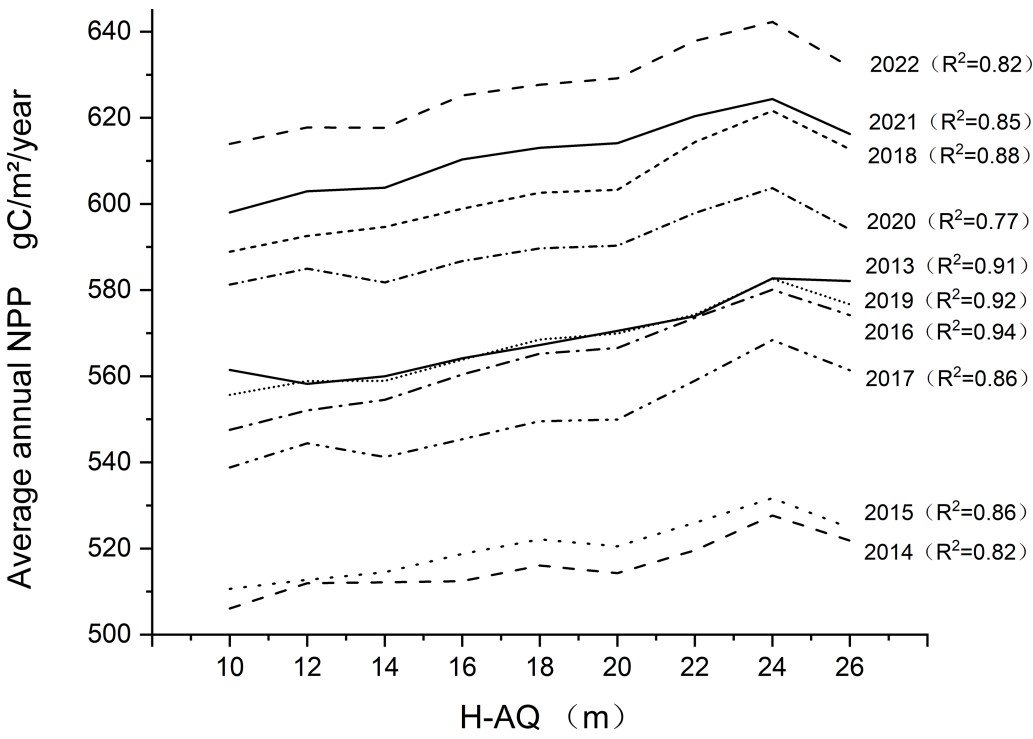

**Figure 7  Annual NPP relationship with H-AQ.** The curve in the diagram represents the changing trend of annual average NPP under each level of H-AQ, and the relationship between NPP and H-AQ is analyzed by linear regression. The right side of the graph shows the year corresponding to the curve and the determining coefficient $R^2$ of linear fitting. The unit of NPP is gC/m²/year.

### Different driving factors affecting NPP value
#### Correlation analysis between site class and NPP value

To verify the correlation between site class and NPP value, we drew a curve of the relationship between site class and average annual NPP value from 2013 to 2022 (Fig. 7). The results show that there is a good linear relationship between annual site classes and the average annual NPP value. Among them, the year with the highest determination coefficient ($R^2$) is 2016, at 0.9401, while the lowest year is 2020, at 0.7657. Average determination coefficient ($R^2$) is 0.86275. As can be seen from Table 6, there is a strong linear relationship between NPP and H-AQ, with the slope ranging from 10.5668 to 19.5745, with the highest slope in 2016 and the lowest in 2014. We also calculated the Spearman's rank correlation coefficient and Kendall's correlation coefficient for each year. The results are presented in Table 7. It can be observed from Table 7 that there was a positive correlation between NPP and H-AQ; the value of NPP increased with the promotion of H-AQ. The above results indicate that the NPP value of the *Larix kaempferi* plantation in Dandong City was significantly correlated with site class.

As can be seen from the linear fitting relationship between NPP and H-AQ in the last ten years (Fig. 8), the results show that its determination coefficient ($R^2$) is 0.92006, indicating

**Table 6 Table of parameters of the relationship between NPP and H-AQ for the year.** Prob > |t| indicates the significance of Pearson correlation coefficient.

| Age | Parameter | Value | Prob > |t| |
|---|---|---|---|
| 2013a | Intercept | 5403.7360 | 1.63213E−13 |
| 2013a | Slope | 15.8647 | 7.89518E−5 |
| 2014a | Intercept | 4967.6172 | 2.34027E−13 |
| 2014a | Slope | 10.5668 | 7.905E−4 |
| 2015a | Intercept | 4994.4732 | 1.61517E−13 |
| 2015a | Slope | 11.5183 | 3.54753E−4 |
| 2016a | Intercept | 5285.6117 | 1.46458E−13 |
| 2016a | Slope | 19.5745 | 1.56785E−5 |
| 2017a | Intercept | 5205.7895 | 1.68817E−12 |
| 2017a | Slope | 16.8432 | 3.47941E−4 |
| 2018a | Intercept | 5693.6831 | 8.09517E−13 |
| 2018a | Slope | 18.8428 | 1.57789E−4 |
| 2019a | Intercept | 5388.7499 | 8.33928E−14 |
| 2019a | Slope | 16.0164 | 3.92166E−5 |
| 2020a | Intercept | 5685.6829 | 6.67134E−13 |
| 2020a | Slope | 11.9316 | 0.00201 |
| 2021a | Intercept | 5853.6045 | 2.76978E−13 |
| 2021a | Slope | 14.5043 | 3.66374E−4 |
| 2022a | Intercept | 5986.2488 | 9.29418E−13 |
| 2022a | Slope | 15.7865 | 7.11769E−4 |

**Table 7 Annual NPP and H-AQ correlation analysis.**

|  | 2013 | 2014 | 2015 | 2016 | 2017 |
|---|---|---|---|---|---|
| Spearman | 0.12099[*] | 0.04030[*] | 0.05904[*] | 0.11995[*] | 0.08439[*] |
| Kendall | 0.08661[*] | 0.02854[*] | 0.04191[*] | 0.08527[*] | 0.05981[*] |
|  | 2018 | 2019 | 2020 | 2021 | 2022 |
| Spearman | 0.11895[*] | 0.08701[*] | 0.08545[*] | 0.10476[*] | 0.11079[*] |
| Kendall | 0.08439[*] | 0.06191[*] | 0.06063[*] | 0.07460[*] | 0.07890[*] |

**Notes.**
Two-tailed significance test was used.
[*]The correlation is significant at the level of 0.05.

a highly significant linear correlation between NPP and H-AQ. When H-AQ increased by 2, the average NPP value in recent ten years increases by about 2.731 gC/m$^2$/year.

### NPP resolution of under different site quality

The NPP quantity distribution of high-quality, average-quality, and inferior sites is shown in Fig. 9. The R of the NPP values for forest land with inferior quality and forest land with average quality, and forest land with average quality and high quality was calculated using Eq. (11).

According to the results in Table 8, there was a separation trend in the NPP value of forest land with different site qualities, but the separation trend was weak. Moreover, the

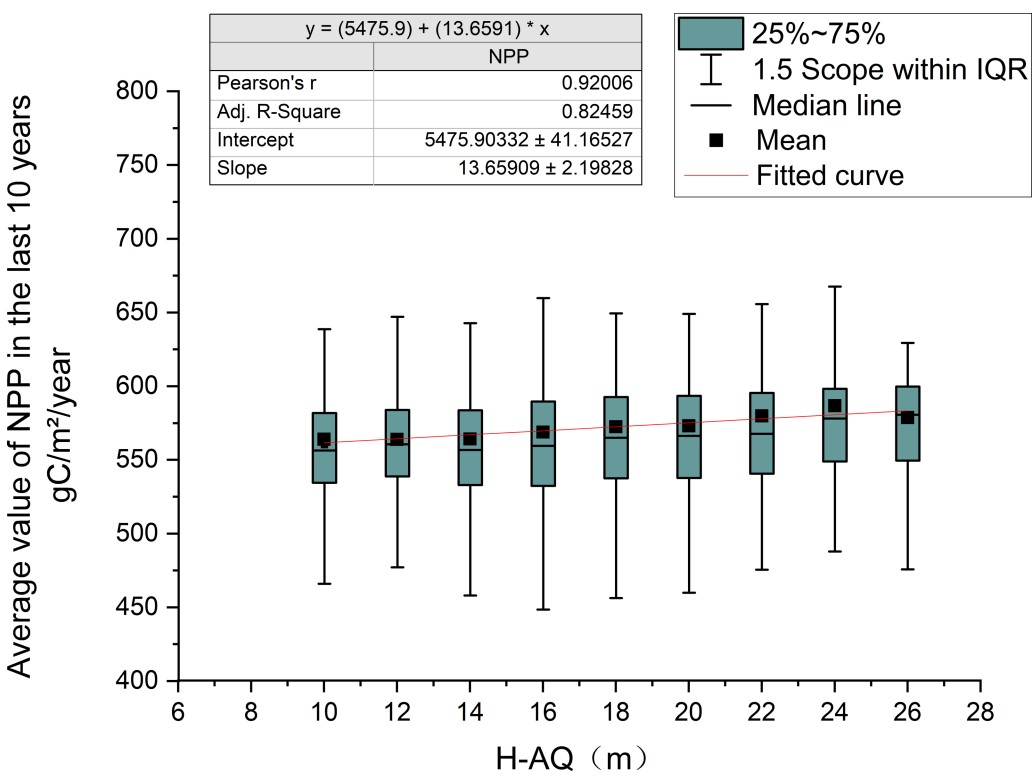

**Figure 8   The relationship between NPP and H-AQ in the past ten years.** The table in the upper left corner shows the linear regression results between the ten-year average NPP and H-AQ, and the red line in the figure indicates the fitting curve. The box plot displays the distribution of the ten-year average NPP under different H-AQ levels.

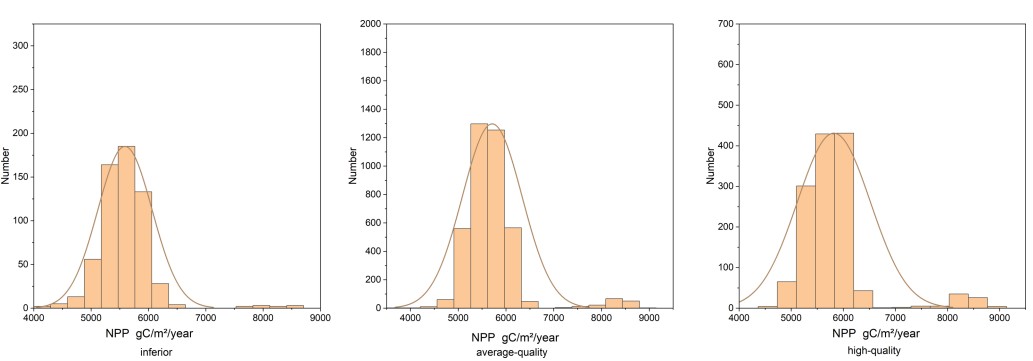

**Figure 9   NPP distribution in inferior, average-quality, and high-quality sites.** Lands with an exponential order of 10~14 are classified as inferior sites, lands with 16~20 as average-quality sites, and lands with 22~26 as high-quality sites.

**Table 8  Resolution.** Lands with an exponential order of 10~14 are classified as inferior sites, lands with 16~20 as average-quality sites, and lands with 22~26 as high-quality sites.

| R Inferior and average-quality | R average-quality and high-quality |
| --- | --- |
| 0.057666 | 0.068485 |

forest land with better site quality showed an overall increasing trend compared with the forest land with poorer site quality.

### Effect of stand age on NPP value

According to correlation analysis between the H-AQ and NPP values, there is a significant positive correlation between the NPP value and site classes in Dandong City. However, the variance of NPP value was relatively large under each site class, despite an overall growth trend (Fig. 8). According to Fig. 10 which shows the relationship between age class and average annual NPP, it can be observed that the maximum NPP values of *Larix kaempferi* in Dandong City, Liaoning Province in 2013 all appears in II (10-20 age class). By observing the age class-NPP curve of *Larix kaempferi* in Dandong City, Liaoning Province in Fig. 5, it can be seen that the rapid growth period occurs between 5 and 20 years. At this stage, the growth rate of *Larix kaempferi* is very fast. Therefore, the average NPP value of forest land of corresponding age class is the highest. This observation is also supported by Fig. 10. In addition, the age classes III and IV show low NPP values, which indicates that the growth of trees enters the mature stage and therefore the growth activity decreases (Fig. 10). Therefore, our observations were consistent with the growth characteristics of *Larix kaempferi,* showing that there is rapid growth between 5 and 20 years, after which trees grow slowly.

## DISCUSSION

SI models developed using the GADA are preferred in many countries with developed forestry management (*Aydin, Turan & Klaus, 2018*). Various studies have been done on SI modeling with GADA (*Adan et al., 2022*; *Guerra et al., 2021*; *Cieszewski, 2002*; *Cao & Sun, 2017*; *Richards, 1959*; *Arias-Rodil et al., 2014*; *Yang et al., 2021*; *Aydin, Turan & Klaus, 2018*). The dynamic SI equations acquired using GADA produced successful results consistent with the expected growth laws and age-dominant height relationships. The SI model can be developed from different data sources, such as stem analysis, repeated data of permanent sample plots established for experimental purposes, or survey data in the context of the National Forest Inventory (*Scolforo et al., 2016*). The accuracy of height growth data varies, and the stem analysis data is usually more accurate than that of permanent sample plot data (*García, 2018*). However, experimental sites are often built on well-managed stands, and permanent sample plots in the National Forest Inventory usually represent large areas of growth conditions. In this study, the average height is used to replace the height of dominant trees, and the mean height-age model is derived. The derived model has advantages (*Cieszewski & Bailey, 2000*) such as age invariance and pleomorphism. Since the basic model derived from GADA is more complex than from ADA, inventory for forest planning and design replaces commonly used stem analysis data with the average age and

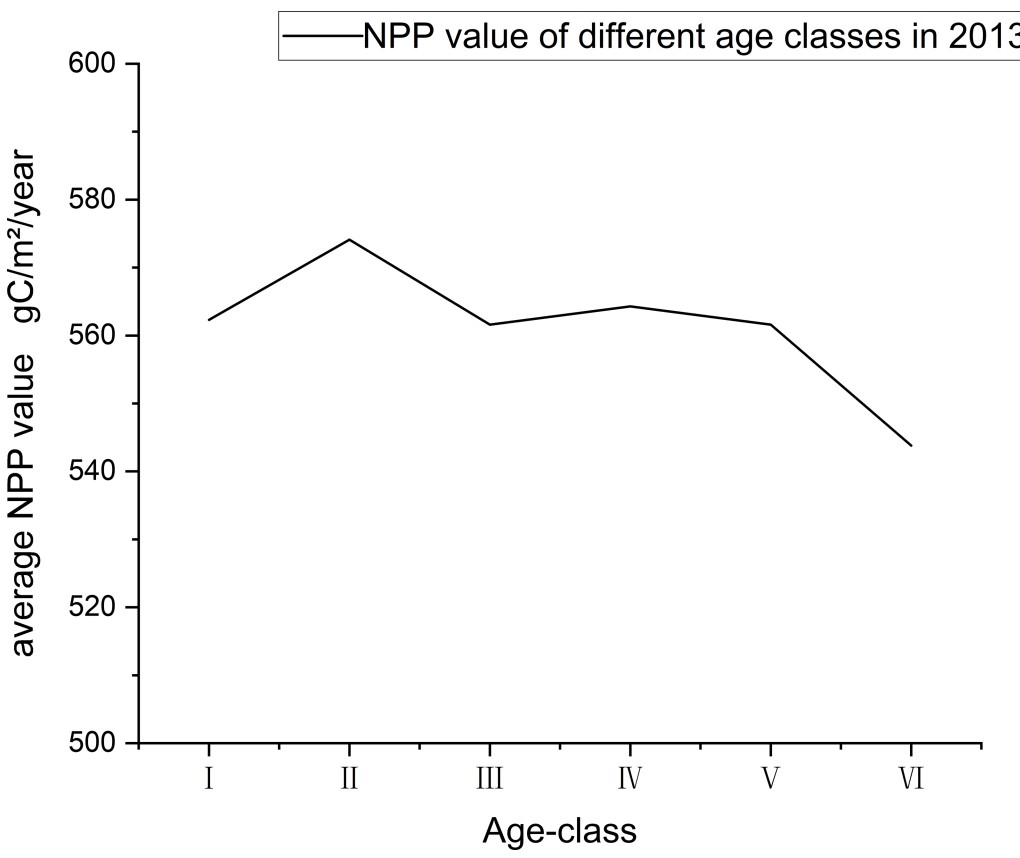

**Figure 10 The relationship between NPP and age class in 2013.** The age classes are as follows: 0–10 years old is I, 10–20 years old is II, 20–30 years old is III, 30–40 years old is IV, 40–50 years old is V, and over 50 years old is VI.

average tree height of the sample plot, resulting in a larger range and quantity of differences in forest stand characteristics and site conditions and higher data variability. Therefore, some basic models using GADA may result in poor model fitting. However, the mean height-age model derived from GADA is still superior to that from ADA through statistical and graphical analysis. Moreover, the research results of *Sharma et al. (2011)* show that some stem analysis data can be further added to the National Forest Inventory to improve accuracy.

The results of the correlation analysis between site class and NPP value show that there is a positive correlation between H-AQ and NPP. The H-AQ comprehensively considers the influence of stable factors (such as climate and topography), comparatively stable factors (such as soil texture and hydrology) and unstable factors (such as pH value, moisture and nutrient content) (*Cieszewski & Bailey, 2000*; *Farrelly, Ni Dhubhain & Nieuwenhuis, 2011*). The NPP value in forest land is affected by climate, topography, soil, water and nutrients (*Chen, 2019*; *Adams, White & Lenton, 2004*). Therefore, the convergence between the H-AQ and NPP can be understood as that they both considered factors related to forest land and forest land productivity when evaluating forest land quality. In this study,

NPP increased with the increase of site quality, and the growth rates were different in different years with an unobvious interannual change. The highest growth was in 2016: NPP increased by about 3.914 gC/m$^2$/year for every 2 increases in H-AQ; the lowest increase in NPP was in 2014: NPP increased by about 2.113 gC/m$^2$/year for every 2 increased in H-AQ; and for every 2 increases in H-AQ in the past ten years, the average NPP increased by about 2.731 gC/m$^2$/year. *Ji et al. (2020)* shows that the change of forest net productivity in China is mainly due to the change of precipitation, followed by a change in temperature. Therefore, the obvious interannual change of NPP value may be related to climatic factors such as precipitation and temperature.

In addition to the environmental factors such as climate, topography, soil, and water, the NPP value was also affected by other factors including the ecological attributes of trees in forest land. From the results of the NPP values under different site classes, it can be concluded that the NPP values in the same site quality are very different, even though the NPP values under different site qualities are separable. This indicates that NPP values of forest land are also affected by other factors besides site quality-related factors. When studying ecosystem functions such as carbon stock and NPP in temperate forest ecosystems, the effects of stand density and species richness should be considered. *Cai et al. (2016)* showed that forest carbon stock and aboveground NPP are affected by stand density, species richness, and stand succession. *Wang et al. (2011)* showed that the NPP value was affected by tree age. The NPP value first increased and then decreased with an increase of age. The forest with higher yield shows earlier growth peak and steeper growth decline. This is because the NPP value was affected by other factors besides site class. Many studies have found that in addition to site quality-related factors, stand age will also have an impact on NPP. It should be noted that the forest land data in this study was based on the forest resource inventory data for management in 2011, the NPP in 2013 was used as an example for analysis, and ten years is taken as an age class. The curve between NPP and stand age in this study also shows that the NPP value is affected by age; the NPP value is the highest in the middle and half-mature age of *Larix kaempferi*. Since the NPP value of forest land is also affected by other non-site factors, the NPP value of sample plots in the same site class is quite different. At the same time, the results of Figs. 10 and 3 show that *Larix kaempferi* grows vigorously during young age in Dandong City, Liaoning Province; and it can be seen from Fig. 3 that its growth rate was the highest between years 5–20. During the high-speed growth period (5–20 years) (*Kim et al., 2013*), the corresponding NPP value also reached a high level (Fig. 10). This indicates that during this period, the trees are in rapid growth and therefore the fixed NPP value for tree growth is also high. It has been shown that when *Larix kaempferi* enters the overmature stage, the growth trend of tree height will tend to stagnate (Fig. 3). Correspondingly, in Fig. 10, the NPP value of this period is also low. This indicates that, after 50 years of growth, the growth rate of *Larix kaempferi* slows. This is consistent with the results of *Jiang (2004)*. Therefore, it is clear that young and half-mature *Larix kaempferi* grows most vigorously between the ages of 5–20; however, growth rate gradually slows down after the tree age exceeded 50 years. According to the current carbon sequestration afforestation plan, *Larix kaempferi* is a suitable tree species in Dandong City. However, choosing a suitable site is very important for this tree

species. The carbon sequestration effect of newly planted afforestation is faster when the trees are younger, but it will gradually weaken with the growth of tree age. Therefore, it is particularly important to cut and renew the stand (*Pauls et al., 2024*). In addition, replacing some fossil fuels with wood can also help to alleviate climate change. The dominant height is used in the mean height-age model because it can more accurately represent the potential height growth of trees in the stand for the following reasons: it is relatively easy to estimate in the field, it is related to mass production, it does not depend on afforestation, and its value will not change during harvesting (at least during low thinning). By using the dominant height, the position index model focuses on the trees that best represent the growth potential of the site (*Teixeira et al., 2023*).  However, in this study, we used the average height of the stand instead of the dominant height to calculate the position index. Although this method may affect the accuracy of the position index, it is difficult to obtain the dominant tree height of a large number of woodlands in a large area. In this study, the relationship between NPP and site quality over a large range of woodland was explored to provide a new perspective for forest carbon sink estimation. Therefore, although using the average stand height as the basis may bring some influence, it provides a practical starting point for us to analyze the relationship between NPP and site quality, which is expected to promote the development of forest ecology and carbon cycle research. In the future research plan, we will consider using the height of dominance tree and more accurate NPP data to discuss the interaction between them more deeply. This improved method will help us understand the growth potential of forest and its role in the global carbon cycle more comprehensively, thus providing a more scientific and accurate basis for forest management and carbon sink estimation.

NPP is highly complex and is affected by more factors than just site quality. Here, we attempted to optimize the model and to include some new mixed model methods (*Protazio et al., 2022*). Some scholars have inserted climate factors into the GADA equation to improve the fitting degree of the model (*Sharma et al., 2011*). Future research should include climate factors in the parameterization method and NPP related factors, such as density, in the GADA method to improve the accuracy of the position index model. These changes may better establish the correlation between NPP and site quality.

## CONCLUSIONS

This study utilized the GADA method to construct a mean height-age model and compared it with the model developed by the ADA method. Unlike traditional modeling methods that mainly rely on analytic tree data, this research used the Forestry Resources Second Class Survey to develop the mean height-age model, demonstrating the significant statistical advantages of the GADA method. This study also validated the effectiveness of the Second Class Survey in fitting the mean height-age model as well as the correlation between H-AQ and NPP. We found that the higher the H-AQ, the more useful its carbon sequestration capacity. The results showed that an increase of two classes corresponded with an elevated H-AQ and the average NPP increased about 2.731 gC/m$^2$/year over almost a decade. This finding is of crucial importance for understanding the carbon sequestration potential

of forests. This study also can provide theoretical support for the estimation of carbon sinks in *Larix kaempferi* forests on a large scale. By analyzing the relationship between different site qualities and NPP values and establishing a linear relationship between H-AQ and NPP, we can assess the growth potential and carbon sink capacity of *Larix kaempferi* under different stand conditions. Thus, it provides important guidance for silvicultural planning and forest carbon sink management of *Larix kaempferi*. It will also help promote the sustainable utilization of forest resources and ecological conservation.

## ACKNOWLEDGEMENTS

We are grateful to the members of forest management department for helpful discussions and comments.

### Funding
This work was supported by The Liaoning Provincial Natural Science Foundation of China (2022-MS-260). The funders had no role in study design, data collection and analysis, decision to publish, or preparation of the manuscript.

### Grant Disclosures
The following grant information was disclosed by the authors:
The Liaoning Provincial Natural Science Foundation of China: 2022-MS-260.

### Competing Interests
The authors declare there are no competing interests.

### Author Contributions
- Wenlong Chang conceived and designed the experiments, authored or reviewed drafts of the article, and approved the final draft.
- JingHao Li analyzed the data, prepared figures and/or tables, and approved the final draft.
- Jinwei Wu performed the experiments, prepared figures and/or tables, and approved the final draft.
- Jian Zhang analyzed the data, prepared figures and/or tables, and approved the final draft.
- Yang Yu analyzed the data, prepared figures and/or tables, and approved the final draft.
- Huiwen Sun conceived and designed the experiments, prepared figures and/or tables, and approved the final draft.
- Yibo Wen conceived and designed the experiments, prepared figures and/or tables, authored or reviewed drafts of the article, and approved the final draft.

### Data Availability
The data, including the required tree height data and Npp data, is available at figshare: wu, wujinwei (2024). Study on the relationship between Net primary productivity and site quality in Japanese larch plantations in mountainous areas of eastern Liaoning. figshare. Dataset. https://doi.org/10.6084/m9.figshare.25859347.v1.

## Supplemental Information

Supplemental information for this article can be found online at http://dx.doi.org/10.7717/peerj.17820#supplemental-information.

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
