# Peer review of "Study on the relationship between net primary productivity and site quality in Japanese larch plantations in mountainous areas of eastern Liaoning"

_PeerJ, doi:10.7717/peerj.17820_

## Round 0.1 · original submission · Major Revisions

The aim of this paper is of great importance because forest productivity is the most important factor in the implementation of forest management strategies. However, there are still a number of improvements that the reviewers have listed. The manuscript is hard to understand and there is a disorder for references.
It is also very important to mention in the introduction and discussion that there are novel hybrid models which are easier to apply than the other models (such as GADA and ADA) and have a lower data collection and computational cost (Protazio et al. Dynamical Model Based on the Chapman-Richards Growth Equation for Fitting Growth Curves for Four Pine Species in Northern Mexico. Forests. 2022; 13(11):1866. https://doi.org/10.3390/f13111866). Please check whether one of these new hybrid models (e.g. the Hybrid Chapman-Richards Model (CR-H)) performs better than the corresponding best model (GADA) you have found.

Reviewer 1 ·

Basic reporting

This paper represents an interesting contribution to the knowledge of Japanese larch plantations, which is important for managing forest plantations of dioecious coniferous trees, and for understanding the factors limiting AGB stocks in forest ecosystems, and its role of forests in the context of climate change. The authors pursue the following objective: to understand the correlation between the site class and carbon sink potential of Larix kaempferi plantations. The objective is potentially advancing scientific knowledge by developing new information for these forest plantations, which apparently, is not covered by existing research. Therefore, I consider that it can be published after taking into account the suggestions included in the attached file.
The manuscript is well written, although some parts of the text (comments directly in the manuscript) need to be improved, due to the identification of incomplete sentences, or the use of "filler words", such as "Among them", which appears at least 15 times throughout the manuscript; many words that should be written in the plural because of the context of the sentence are written in the singular; also evident is the lack of use of articles, such as “the”. The manuscript should be reviewed by a fluent English speaker, as it contains grammatical and typographical errors that make it difficult to understand some sentences.
The introduction presents sufficient, though not entirely relevant, background information on studies related to the subject of the article. The justification of the study is not clear, nor is the generation of new knowledge.

Experimental design

Methodology is clear, although not sufficiently detailed; however, the methods used, calculations performed and statistical tests support the results obtained.

The most important weakness I found in this contribution is that it does not use the dominant height of the stands to adjust the site index models, thus violating the site index principle. Dominant height is used in site index models because it provides a more accurate representation of the potential height growth of trees in a stand for the following reasons: it is relatively easy to estimate in the field, it is related to volume production, it does not depend on silviculture and its value does not change when cutting (at least in low thinning). By using dominant height, the site index model focuses on the trees that are most representative of the site's growth potential.

After reading the methodology several times, I could not find the way in which the authors measured the dominant height of the stand, but in the results (tables xx) the dominant height is reported. What the authors do mention is that they used the average height (line xx). The fact that the authors used mean stand height means that the models developed invalid for the objectives of the study.

Validity of the findings

The results are clear but not concise; tables are not sufficiently explained, and not all tables (table 7 and 8) are mentioned in the results or elsewhere in the manuscript. The wording of the discussion is not clear, since it includes complete paragraphs that belong to the results section. Finally, there are no clear and concrete conclusions, rather the authors repeat the results. Although the results obtained are justified, the implications of these results in the context of climate change are not highlighted. This is another weaknesses of the article that needs to be addressed, as the authors claim that this is the contribution of their research.

Additional comments

See attachment.

Annotated reviews are not available for download in order to protect the identity of reviewers who chose to remain anonymous.

Reviewer 2 ·

Basic reporting

The purpose of this paper is of great significance and clearly addressed. The forest productivity is the most important aspect when the forest management strategies taking. I think readers will be interested in this study. however, it has some point should be improved before publish in our journal. I recommend this manuscript to be published in your journal after major revision.

The submission was a little not self-contained, and not represent an appropriate unit of publication. it should be rewriten in the manuscript.

the redults was a little not relevant to the hypothesis.

Experimental design

1. The Introduction need further improvement and supplement.

2. Some abbreviations are not required in the Abstract, the full name should be used when used in the first time.

3. The Conclusion should be strengthened to the focus of the topic of your work.

4. Figure 1 should add the coordinate of latitude and longitude.

5. The name of figure 3 should describe the diagram in more detail and introduce each piece of information in the diagram so that the reader can better understand what the diagram expresses.

6. the methods is the not the best for the NPP and sites quality specifically for the conifer species

Validity of the findings

7. The equation should be rewritten by using the Math type software.

8. Authors should declare the study site had not been effected by human disturbance.

9. It also needs to further illustrate the limitations and deficiencies of this research.


10. the conclusion should be appropriately stated, should be comected to the original question investigated, and should be limited to those supported by the results.therefore, the conclusion of the manuscript shoul be polished and improved.

Additional comments

1. Figure 6 is too blur to be adjusted,the corresponding curves for 2013, 2016 and 2019 are not clear. The drawing should be revised.

·

Basic reporting

1. Basic reporting
The manuscript is well written, but it is hard to follow the objectives, methods, results and conclusions. Some paragraphs in Results and Discussion sections seem methods.
The authors kindly suggested to use “SI” instead “site index” in the entire manuscript. Sometime SI is used and sometime site index. Please, use ADA or GADA after first mentioning.
The GADA methodology should be clarified and improved in the entire manuscript.
Some Tables are missed for the main text.
Conclusion should be rewritten in a single paragraph. Please avoid to add mathematical formulas in this section.
Authors should explain and justify the ADA and GADA models. The E1 is an ADA model but the solution is false.
Please, improve the Tables and Figures.
Additionally, all information presented in results should be added in Materials and methods.
Please, all tables and Figures should be added in the Results section.

Experimental design

2. Experimental design
The experimental design is interesting. But the used dataset to fit the ADA and GADA equations comes from forest inventory. Authors should explain in detail. Hot the ADA and GADA equations were fitted to forest inventory dataset.
If the h1 parameter was fixed for a specific age, authors should explain and justify. The results seem as a fitted curve guide equation.

Validity of the findings

In this section is hard to understand the manuscript. The objectives does not match methods, results, and conclusions. The Study is hard to follows and the cites literature is disorganized.

Additional comments

4. General comments
The comments for authors were divided into two sections; general comments and specific comments.
General comments.
The using of references should follow a congruent style according to the Authors guide and how they should be shown in the manuscript. Please. Avoid spaces in before and after all references
Specific comments and suggestions.
Title section
Looks good.
Abstract section
Please, improve this section and use past tense for main sentences.
L14. Please use italic font for species name and do the same in the entire manuscript (MS).
L18—19. Please, use the same order for growth equations. This should match the order in in L211—216.
L23. “was”.
L24. “were”.
L25. “was”.
L26. Please, introduce the E1 equation.
Please, use SI for site index.
L29. “was”.
L30. What is NPP?
L31. Please, explain MOD17A3HGF.
L38 “Larix kaempferi”.
Introduction
L41. “et all” or “et al”?
L41. Please avoid spaces before and after reference. The same in L48 and in the entire MS.
L51. “al”.
L55. Please, improve reference. Do the same in the entire MS.
L64. I think is “et al” instead of “et all”.
L82-83. “ The Modern Resolution Imaging Spectradiometer (MODIS)…”.
L86. Improve references. Please, add a comma or point.
L93. Please adda the reference at the end of sentence.
L94—117. Please, improve all reference.
L114. “index age or base age”.
L114. “SI”.
L115. “SI”. Please, do the same in the entire MS.
L117. “SI”. Please, improve the reference.
L119—120. Please, improve the paragraph and each sentence.
L121. “SI”. And do the same in the entire MS.
L136 and L138. “were”.
Materials & methods
L142. “subsection”?
L151. Please, rewrite this sentence.
L152. “… C, and North …”.
L167—168. Please, explain in detail.
L169. “were”.
L170 “were”.
L176 “ 500 m”.
L178. What is HDF?, please, explain.
L185. “were”.
L189. “ADA”.
L190. “ADA”.
L192. “SI”.
L192—194. “(2) …; (2), …, (3)”.
194, “GADA”.
L196—197. Rewrite.
L207—208. Please, follow the same order in equations. Please avoid #.
L208. “[56]?”.
L209 “SI”.
L221. “2.3.2?”.
L222. Improve the first sentence.
L222. “ADA” instead of "algebraic difference approach”.
L228. “SI”.
L234. “t0 and h0”. If you are talking about X0, you should use t0 and h0.
L240. “SI”.
Please, improve all GADA formulations. You should use 0 and 1 or 1 and 2, but not 0 and 2 stages.
Please, avoid #.
L245. Please, use h0, t0 and X0. Also, alpha’s parameter could be used.
L256—257. This is false. Each parameter a, b, and c has a specific solution and X0 has a different solution.
L257. “[53]?”.
The solution for X for E0 in Table 3 is wrong. That is the solution for exp(X0). Neither ADA nor GADA.
L267. BIAS and MAE are the same. Please use only one. I kindly recommend authors to use BIAS.
L274. Please, use “1” instead of “n” for this formulation.
L281. Please, check the reference for R o RStudio.
L298. Please add a reference.
Authors should explain in detail how the GADA models were fitted. They use nls and the h0 parameter was not explained. The GADA models were fitted by guide curve and the results showed that. If it is true, base growth models could be used or ADA models.
Authors should rewrite the GADA theory and its formulation.
L291. This was not used in the Results section.
Please, add more detail about NPP was obtained for MODIS. How was this variable ranged?
L295—296. Please, rewrite these sentences.
Results
L304. “were”.
L304. “SI”.
L326. “(1)”.
L327. “(2)”.
L327. “(3)”.
L327. “(4)”.
L329. “SI”.
L337. “SI”. The same for L338.
The equation E0 has two parameters and the equation E1 has three parameters. Please, remove all GADA equations which were not convergence in fitting process. Please, just include the GADA equations with convergence.
Please, include the comparation between GADA models with three parameters. The ADA equation has only two parameters. They are comparable.
L372—392. Please remove this information. This is not necessary.
L394. Formulas 14 and 15 are missed. Also, remove this information.
L395—400. Please. Move to materials and methods section.
L417. Please. Improve Figure 6.
L426—431. Please. Move to materials and methods section.
L437—441. “sep-aration?”.
L442. “3.3.1?”.
Discussion
L462. “SI”.
L462—467. Please, improve this paragraph and all references.
L467. “SI”. The same in L476.
L488—507. This information is not relevant for the study. Please, remove or rewrite it.
All Tables and Figures should be included in the Results section.
Conclusions
Please, improve this section in a single paragraph (Less than 250 words). Please, avoid formulas.
Please use GADA instead of Algebraic Difference Approach.
This section should be rewritten.
Figures and Tables
The information of Figure 7 should be added in the material and methods section.
Please, improve Table 1. Please, add a description below table for each variable.
The solution for X in E0 is wrong. This solution is for exp(X0). I think you are talking about the “a” parameter solution of base growth model. Please, include only equation with convergence in fitting process.
Why E2 did not converge in the fitting process? This equation is so popular and well performance could be shown. Is this about the used dataset?, please, explain in detail.
Only include E0, E1. The rest of GADA equations performed poorly.
The BIAS and MAE should be the same. Just MAE in positive values. Please, use only the BIAS statistic.
Please use “<0.00001” for p-value instead of “<2e-16”.
Information of Table 6 should be added in the materials and methods section.
Table 7 is missed in MS.
Table 8 is missed in MS.

---

## Round 0.2 · Major Revisions

The manuscript has been significantly improved. However, there are still some suggestions for improvement from 2 reviewers, especially in the methods section. The selection criteria for choosing specific models for comparison could be further elaborated to demonstrate the rigor of the experimental design.
There are methodological areas where more specificity is needed, particularly in the data preprocessing steps and the criteria for model selection and evaluation. Providing detailed protocols or code used in the analysis would significantly improve reproducibility.

As in my first response as editor, I miss the mention in the introduction and discussion that there are novel hybrid models that are easier to apply than the other models (such as GADA and ADA) and have a lower data collection and computational cost (Protazio et al. Dynamical Model Based on the Chapman-Richards Growth Equation for Fitting Growth Curves for Four Pine Species in Northern Mexico. Forests. 2022; 13(11):1866. https://doi.org/10.3390/f13111866). Please check whether one of these new hybrid models (e.g. the hybrid Chapman-Richards model (CR-H)) performs better than the corresponding best model (GADA) you have found.

Reviewer 1 ·

Basic reporting

The authors have enhanced the experimental methodology outlined in the article while streamlining the discussion section, thus improving the overall coherence and logic of the paper. Additionally, they improved the conclusion to succinctly encapsulate the implications of the findings on aboveground biomass and carbon estimation. Extensive and repetitive sentences have been improved, resulting in a more concise and lucid manuscript. Thorough scrutiny has been applied to the entire document, aiming for linguistic enhancements throughout.

Experimental design

I am concerned that the authors again use mean height instead of dominant height. Using average height instead of dominant height to develop site index curves raises several fundamental concerns in evaluating the quality of a forest site. Here three more arguments supporting the idea that average height should not be used in this assessment:
- Dominant height, reflecting the average height of the tallest trees in a given area, provides a more representative measure of optimal growth conditions within the forest site. These taller trees are often the best indicators of the site's growth potential and prevailing environmental conditions. Using average height may underestimate the true productivity of the site by not adequately capturing the upper canopy.
- Dominant height tends to be a more stable and reliable measure over time compared to average height. This is because taller trees are less influenced by competition with other trees and tend to maintain their dominant position in the canopy over multiple growth cycles. Using average height may introduce greater temporal variability in site index evaluation.
- Dominant height has been established as a more accurate and reliable measure for assessing site productivity and predicting forest yield. Using average height may lead to less accurate estimations of site productivity, which could have significant implications for forest planning and natural resource management.

Since the dominant height is not used, it is not possible to speak of a site index; therefore, the authors should remove this term from the manuscript and emphasize that it is a mean height-age relationship. Similarly, in Figures 3 and 4, the title of the y axis should be changed, i.e., mean height/m instead of dominant height/m.

Validity of the findings

No comments

Additional comments

No comments

Reviewer 2 ·

Basic reporting

1. The manuscript is well-written in professional English, facilitating clear understanding. However, there are instances where the language can be further polished to enhance clarity. For example, the use of technical jargon could be minimized or well-defined for readers not familiar with the specific field of study. It is recommended that the manuscript undergoes proofreading by a fluent English speaker specialized in scientific writing to ensure that the language is clear and unambiguous throughout.
2. The background provided in the introduction section sets a good context for the study, citing relevant literature. Nonetheless, there seems to be a gap in discussing recent advancements in the field, particularly those related to climate change impacts on forest carbon dynamics beyond 2020. Incorporating a more comprehensive review of the latest studies would strengthen the context and relevance of the research.
3. The figures and tables are relevant and support the findings well. However, the raw data's accessibility could be improved. It is encouraged to share the raw data in a publicly accessible data repository with proper metadata, adhering to PeerJ's policy on raw data sharing. This will enhance transparency and allow for reproducibility of the results.

Experimental design

The research question is well-defined, and the study addresses an identifiable knowledge gap. The use of the Generalized Algebraic Difference Approach (GADA) for modeling is adequately justified. However, the selection criteria for choosing the specific models for comparison could be further elaborated to demonstrate the rigor in experimental design.
Methods are described with sufficient detail, but there are areas where more specificity is required, especially in the data preprocessing steps and the criteria for model selection and evaluation. Providing detailed protocols or code used in the analysis would significantly enhance replicability.

Validity of the findings

1. The conclusions drawn are supported by the results; however, they sometimes extend beyond the scope of the presented data. For instance, the implications of the findings for forest management and carbon sequestration strategies are discussed without a detailed analysis of the practical applicability and limitations of the study. The discussion should more critically assess the broader applicability of the SI models developed and their implications under different climatic scenarios.
2. While the statistical analysis appears robust, the interpretation of some results, particularly the correlation between site index (SI) and net primary productivity (NPP), could benefit from a deeper exploration of the ecological and physiological mechanisms underlying these relationships. Moreover, considering the complexity of factors influencing NPP, including but not limited to site quality, a more nuanced discussion on the potential confounders and their impact on the study's findings would be valuable. General Comments

3. The manuscript makes a significant contribution to understanding the relationship between site quality and NPP in Japanese larch plantations. However, to improve the manuscript's impact and reliability, it would be beneficial to address the above points. Specifically, enhancing the clarity and depth of the methods section, providing a more critical analysis in the discussion, and ensuring transparency in data sharing are crucial steps.
- Additionally, exploring the implications of the findings within the broader context of global change, forest management, and conservation strategies in more detail would enhance the manuscript's relevance and applicability.

·

Basic reporting

Study on the relationship between Net primary productivity and site quality in Japanese larch plantations in mountainous areas of eastern Liaoning.
The revised manuscript is well written. Main of suggestions were considered, and authors improved the manuscript.

Experimental design

The explanation is better and mathematical formulations were improved.

Validity of the findings

This section was improved.

Additional comments

General comments.
The revised version of manuscript was improved and it looks better than previous version.
Specific comments and suggestions.
Title section
Looks good.
Abstract section
Looks better. All species names should be write in italic font.
Introduction
Good job.
Materials & methods
Looks better. The not6 converged model were removed.
Results
Looks better. Good job
Discussion
It was improved and looks better.
Conclusions
Thanks. Good job.
Figures and Tables
In general, figures and Tables were improved.

---

## Round 0.3 · Minor Revisions

1. Please write the newly inserted text in a more comprehensible way. 2.Please describe in more detail in the legends of the tables and figures what can be found in the tables and figures. Also state the units of the variables. For example, nobody understands what “H-AQ” is. What are the units of H-AQ and NPP? The figures and tables must be self-explanatory. 3. Figure 7: R2 with only 2 decimals after the point. 4. Please verify the English again.

---

## Round 0.4 · Minor Revisions

The Mansukirpt is now almost finished. But some of the legends in the tables and illustrations are still too inaccurate. Please improve them. The figures and tables must be self-explanatory.

Then please urgently improve the content of the following text: "Some scholars have developed new mixed models based on the GADA method, which have lower data collection and calculation costs, while others have expanded the application scope of the GADA method (Protazio et al,2022). They use the GADA model to predict the annual cumulative resin output of Pinus pinaster Ait (Oscar LA Luis FV & Manuel MP, 2023). However, the modeling data used by GADA to build the SI model mainly include the stem analysis data of dominant trees and the inventory data of fixed sample plots (Souza et al , 2022; Mfilho et al , 2023)." The hybrid models of Protazio et al. work on the basis of ADA, not GADA. Please check. The following sentence ("They use...") does not match the previous sentence.

---

## Round 0.5 · accepted · Accept

I think that the manuscript can now be published.